# Safety and Efficacy of Bispecific Antibodies in Adults with Large B-Cell Lymphomas: A Systematic Review of Clinical Trial Data

**DOI:** 10.3390/ijms25179736

**Published:** 2024-09-09

**Authors:** Elena Bayly-McCredie, Maxine Treisman, Salvatore Fiorenza

**Affiliations:** 1Epworth HealthCare, East Melbourne, VIC 3002, Australia; elena.bayly-mccredie@epworth.org.au (E.B.-M.); maxine.treisman@epworth.org.au (M.T.); 2Faculty of Medicine and Health, University of Sydney, Camperdown, NSW 2006, Australia

**Keywords:** large B-cell lymphoma, diffuse large B-cell lymphoma, bispecific antibody, epcoritamab, glofitamab, mosunetuzumab, odronextamab, plamotamab

## Abstract

Bispecific antibodies (bsAbs) are an emerging therapy in the treatment of large B-cell lymphomas (LBCLs). There is a gap in the research on the safety and efficacy of bsAbs in adults with LBCL, with current research focusing on the wider non-Hodgkin’s lymphoma population. To address this research gap, we conducted a systematic review aiming to evaluate the safety and efficacy outcomes of bsAbs in adults with LBCL. A systematized search was conducted in PubMed, EMBASE, and CENTRAL on 10 April 2024. Interventional clinical trials were eligible for inclusion. Observational studies, reviews, and meta-analyses were excluded. According to the Revised Risk of Bias Assessment Tool for Nonrandomized Studies, the included studies were largely of a high quality for safety outcome reporting, but of mixed quality for efficacy outcome reporting. Due to the heterogeneity of the included studies, the results were discussed as a narrative synthesis. Nineteen early phase studies were evaluated in the final analysis, with a pooled sample size of 1332 patients. Nine bsAbs were investigated across the studies as monotherapy (nine studies) or in combination regimes (10 studies). The rates of cytokine release syndrome were variable, with any grade events ranging from 0 to 72.2%. Infection rates were consistently high across the reporting studies (38–60%). Cytopenias were found to be common, in particular, anemia (4.4–62%), thrombocytopenia (3.3–69%), and neutropenia (4.4–70%). Immune effector cell-associated neurotoxicity syndrome (ICANS) and grade ≥3 adverse events were not commonly reported. Promising efficacy outcomes were reported, with median overall response rates of 95–100% in the front-line and 36–91% in terms of relapsed/refractory disease. The results of this systematic review demonstrate that bsAbs are generally well-tolerated and effective in adults with LBCL. BsAbs appear to have superior tolerability, but inferior efficacy to CAR T-cell therapies in adults with LBCL. Future research on safety and efficacy should focus on evaluating adverse event timing and management, the impact on the patient’s quality of life, the burden on the healthcare system, and overall survival outcomes.

## 1. Introduction

Large B-cell lymphomas (LBCLs) are a heterogeneous group of high-grade tumors within the non-Hodgkin’s lymphoma (NHL) classification [1]. These tumors arise from cells of mature B-cell lineage and are typically characterized by large lymphoid cells forming sheets or clusters [2]. Diffuse large B-cell lymphomas (DLBCLs) are the most common form of LBCL in the Western world and represents 30–40% of cases of NHL globally [1]. The highest proportion of cases are diagnosed in 65–74 year olds, with a slight male predominance [3]. The first-line treatment for LBCL is chemoimmunotherapy consisting of an anthracycline backbone and an antigen-directed monoclonal antibody, with the most widely used regimen being rituximab with cyclophosphamide, doxorubicin, vincristine, and prednisone (R-CHOP) [4]. For the 35–40% of patients with relapsed/refractory disease, current treatments include salvage chemotherapy with autologous stem cell transplant [4], antibody–drug conjugates [5], monoclonal antibodies [6], and chimeric antigen receptor (CAR) T-cell therapy [7]. 

Bispecific antibodies (bsAbs) are an emerging class of therapy in the treatment of adults with LBCL, with at least 16 bsAbs currently undergoing clinical trials in LBCL [8]. BsAbs are an off-the-shelf immunotherapy constructed to target two antigens through two separate arms [8]. By binding with immune effector cells with one antigen-binding region and the tumor-associated antigen with the other antigen-binding region, bsAbs promote immune-mediated immune cell activation and tumor cell death [8]. The commonly targeted tumor-associated antigens (TAA) in lymphoma are CD19 and CD20, both highly expressed on the surface of B-cells. Upon contemporaneous binding of TAA on the tumor cell and an activating CD3 moiety on the T-cell surface, T-cell activation occurs resulting in perforin/granzyme release and subsequent tumor cell lysis [9]. An alternative immune cell target is CD47, an antigen upregulated in B-cell malignancies and other cancers, that inhibits phagocytosis by macrophages and the so-called “Don’t eat me” signals of associated cell types. Co-engaging CD47 with CD19 permits specific phagocytosis of B-cells and also appears to prevent CD19 cluster formation, thereby inhibiting B-cell receptor proliferation [10].

Overt immune activation is a class effect of bsAbs, resulting in characteristic toxicities, such as cytokine release syndrome (CRS), immune effector cell-associated neurotoxicity syndrome (ICANS) and, rarely, immune effector cell-associated hematologic toxicity (ICAHT) [8]. On-target off-tumor toxicities can result in B-cell depletion and subsequent hypogammaglobulinaemia with ensuant risk of infection, further compounded by immunosuppression due to CRS and ICANS [11]. In 2023, the U.S. Food and Drug Authority approved the bsAbs, epcoritamab and glofitamab, for relapsed/refractory LBCL, paving the way for subsequent antibodies and trials [12,13]. The growing number of new drugs and trials mandates a systematic analysis of the efficacy and adverse effects of bsAbs in adults with LBCL.

A preliminary search of the literature shows that current research is focused on the safety and efficacy of bsAbs in the wider NHL population [14,15,16,17]. While patients with LBCL are included in these publications, combining outcomes for indolent and aggressive lymphomas could result in misestimation of the toxicities in LBCL. Patients with LBCL may be at differential risk of adverse events due to rapidly progressive lymphoma, comorbidities, and toxicities from prior therapies [18,19,20]. Additionally, recent publications on lymphoma and other B-cell malignancies have highlighted infections as a significant toxicity risk of bsAbs [21,22]. As infections have been established as an independent predictor for mortality in DLBCL [20], it is important to review the rates of this toxicity in adults with LBCL exposed to bsAbs in comparison to standard-of-care therapies. Further research is needed to address these gaps in the literature on the toxicities of bsAbs in adults with LBCL. 

The aim of this systematic review is to evaluate the safety and efficacy of bsAbs in adults ≥18 years with LBCL. The primary objective is to identify the frequently reported treatment-emergent adverse events (TEAEs) in this population. The secondary objective is to evaluate the efficacy outcomes of bsAbs in this setting. By collating data on adverse events and efficacy outcomes associated with bsAbs in adults with LBCL, any prominent trends can be identified and evaluated. This may also highlight the potential impact of adverse events and any important differences between bsAbs and standard-of-care therapies.

## 2. Materials and Methods

This systematic review was conducted according to the Preferred Reporting Items for Systematic Reviews and Meta-Analyses (PRISMA) 2020 guidelines and checklist [23]. 

### 2.1. Search Strategy

The search strategy was designed in consultation with Patrick Condron, Librarian at the University of Melbourne. The search was performed on 10 April 2024, using key databases reporting clinical trials results. These included PubMed, EMBASE, CENTRAL, and pre-print servers bioRxiv and medRxiv. The MeSH terms and keywords used included “toxicity”, “safety”, “tolerability”, “lymphoma”, “bispecific antibody”, and “T-cell engager”. The search was limited to English language results published between 1 January 2008 and 10 April 2024. A search of the grey literature was also performed through selected conference abstract books to capture data not yet published or indexed in the databases. A targeted search of the conference abstracts was repeated on 18 July 2024 to capture any new study results reported after 10 April 2024. The results of the search strategy were imported into EndNote 21.3 [24]. For the full search strategy, refer to Appendix A.

### 2.2. Eligibility Criteria

Screening was performed using the online systematic review software Covidence [25], according to predetermined eligibility criteria. Duplicates were automatically removed by Covidence. Title and abstract screening were performed by a single author (EBM), while full-text screening was performed by two authors independently (EBM and SF). Disagreements were discussed, to reach a consensus. 

The inclusion criteria included: (1) interventional clinical trials of bsAbs; (2) adults ≥ 18 years with newly diagnosed or relapsed/refractory LBCL; (3) published in English between 1 January 2008 and 10 April 2024; and (4) comprehensive adverse event data reported as TEAEs. The exclusion criteria included: (1) observational studies; (2) meta-analyses; and (3) adverse event data reported only as treatment-related adverse events (TRAEs). TEAEs were chosen for inclusion over TRAEs to reduce the risk of reporting bias by excluding subjective judgements of adverse event causality made in individual studies [26]. To give priority to peer-reviewed publications with comprehensive outcomes reported, conference abstracts were only included where a full-text report had yet to be published for the same clinical trial cohort. To capture the most up-to-date data, only the most recent analysis from the same clinical trial cohort was included.

### 2.3. Data Extraction

The study characteristics and outcome data were extracted into predetermined data collection forms. The primary outcome data were extracted independently by two authors (EBM and MT). Additional information was sought from ClinicalTrials.gov (accessed on 26 May 2024) [27], publications related to the included studies, and by contacting the source authors. There were no assumptions made about missing or unclear information, with missing results documented as not reported (NR). 

The extracted data for the key characteristics and outcomes of the included studies are presented in tabular form (Table 1, Appendix A). The study characteristics included trial design, key eligibility criteria, demographics of enrolled subjects, treatment schedule, median treatment exposure, adverse event mitigation strategies, and adverse event grading criteria used. The adverse event outcomes included CRS, ICANS, infection, fever/pyrexia, fatigue, cytopenias, and other frequently occurring adverse events. These were defined as any grade adverse events reported in >20% of participants to an individual study and grade ≥3 events reported in >5% of participants to an individual study. The efficacy outcomes included the median follow-up (MFU), overall response rate (ORR), complete response rate (CRR), partial response rate (PRR), median duration of the response (DoR), median progression-free survival (PFS), and response criteria used. Data on the adverse event outcomes were sought from the patients included in the safety analysis, while data on the efficacy outcomes were sought from patients included in the efficacy analysis from all the included studies. 

### 2.4. Quality and Risk of Bias Assessment

The Risk of Bias Assessment Tool for Nonrandomized Studies (RoBANS 2) [28] was used to assess the risk of bias in the intervention and outcomes from the included studies. 

### 2.5. Data Synthesis

Due to the heterogeneity of the included studies, the analysis was performed using a narrative synthesis. The safety synthesis was performed by grouping together related outcomes and reviewing the results from all the included studies reporting on each outcome. Within each outcome group, heterogeneity was explored by reviewing the differences between bsAbs in monotherapy or combination therapy, treatment-naïve or relapsed/refractory patients, and CD19 or CD20-targeting antibodies. The efficacy analysis was performed by grouping together front-line therapies and salvage therapies. The effect measures were explored through tabular presentation to display the spread of effects. The median effect was calculated for each outcome per bsAb from all the studies reporting results. Where the results for key outcomes were missing, selective non-reporting was assessed to determine the risk of reporting bias. The overall certainty of the evidence for each outcome was assessed based on the sample size and the quality of the contributing studies. 

## 3. Results

### 3.1. Study Selection

There were 1617 studies identified through the database searches and a further 106 studies identified through grey literature searching. After removing 415 duplicates, title and abstract screening were performed on 1308 studies. By sorting each individual study by the clinical trial identification number, only those studies that reported on the largest, most comprehensive, and recent results from the same dataset were included. Double-blind, full-text screening was performed independently on the remaining 127 studies. There were 19 studies selected for inclusion, according to Figure 1.

### 3.2. Study Characteristics

All of the included studies were phase I and/or II, multicenter, open-label clinical trials [29,30,31,32,33,34,35,36,37,38,39,40,41,42,43,44,45,46,47], as shown in Table 1. There were nine different bsAbs being investigated across the 19 included studies. The CD20/CD3-targeting bsAbs included odronextamab [32], glofitamab [33,34,35,36], epcoritamab [37,38,39,40,41,42], plamotamab [43], GB261 [44], and mosunetuzumab [45,46,47]. The CD19/CD3-targeting bsAbs included blinatumomab [29] and AZD0486 [30]. TG-1801 was the only CD19/CD47 bsAb [31]. Nine studies investigated bsAbs as monotherapy [29,30,32,36,41,42,43,44,46], while 10 studies investigated bsAbs in combination with other therapies [31,33,34,35,37,38,39,40,45,47].

The demographic spread of the participants enrolled was comparable across the 19 studies [29,30,31,32,33,34,35,36,37,38,39,40,41,42,43,44,45,46,47]. The number of participants enrolled ranged from 10 to 218, with a pooled sample size of 1332 participants. The median age of the enrolled participants ranged from 58 to 79 years [29,30,31,32,33,34,35,36,37,38,39,40,41,42,43,44,45,46,47]. All of the studies reporting the gender mix enrolled a majority of male participants, ranging from 52.2% to 70% [29,30,31,32,34,35,36,40,41,43,44,45,46,47]. The reported median exposure to the bsAb ranged from 46.8 days to 8.7 months [29,31,32,36,37,41,45,46,47]. Four studies investigated bsAbs in treatment-naïve patients [33,37,38,45], while the remaining 15 studies were investigating bsAbs in relapsed/refractory disease [29,30,31,32,34,35,36,39,40,41,42,43,44,46,47]. For patients with relapsed/refractory disease, the median number of prior lines of lymphoma therapy ranged from one to four [29,30,31,32,34,35,36,39,40,41,42,43,44,46,47].

There was variability in the adverse event grading criteria used across the included studies (see Appendix A). In 10 of the studies [30,32,34,36,41,42,44,45,46,47], CRS was graded according to Lee et al. (2014) [48] or the American Society for Transplantation and Cellular Therapy (ASTCT) consensus grading [49]. The CRS grading criteria were not reported in seven studies [29,33,35,37,38,39,40,43]. ICANS grading was less consistently reported. Three studies used ASTCT consensus grading [30,41,42], three studies used a list of ICANS-like neurological symptoms [32,36,45], and the remaining 13 studies did not report any specific ICANS grading method [29,31,33,34,35,37,38,39,40,43,44,46,47]. In 12 of the studies [29,30,31,32,35,36,41,42,43,45,46,47], all other adverse events were graded according to the National Cancer Institute (NCI) Common Terminology Criteria for Adverse Events (CTCAE) [50]. The version of NCI CTCAE grading used varied between the included studies, with the older studies using version 4.0 or 4.03 and the more recent studies using version 5.0. Seven studies, all of which were conference abstracts, did not report how adverse events were graded [33,34,37,38,39,40,44].

**Table 1 ijms-25-09736-t001:** Key study design characteristics of the included studies.

Citation	Intervention	Trial Design	Key Eligibility Criteria	Subject Demographics
Viardot et al., 2016 [29]	Blinatumomab (CD19/CD3) as monotherapy	Phase II Multicenter Open label Single arm Dose escalation	Relapsed or refractory to ≥1 prior line and ineligible or failed autologous HSCT ECOG performance status 0–2 Life expectancy ≥ 12 weeks	23 patients with DLBCL (cohort I and III combined) Median age 66 years 52.2% male participants Median three prior lines of therapy
Gaballa et al., 2023 [30]	AZD0486 (CD19/CD3) as monotherapy	Phase I Multicenter Open label Single arm Dose escalation	Relapsed or refractory CD19+ B-NHL failing ≥2 prior lines of therapy, including prior CD19 CAR T and/or CD20 TCE	62 patients with B-cell NHL (including 27 patients with DLBCL) Median age 68 years 58% male participants Median three prior lines of therapy
Hawkes et al., 2022 [31,51]	TG-1801 (CD19/CD47) in combination with ublituximab	Phase I Multicenter Open label Multi-arm Dose escalation	Relapsed or refractory B-cell lymphoma after ≥1 prior line of therapy ECOG performance status 0–2	16 patients with B-Cell NHL (including nine patients with DLBCL) Median age 71 years 68.8% male participants Median three prior lines of therapy
Bannerji et al., 2022 [32]	Odronextamab (C20/CD3) as monotherapy	Phase I Multicenter Open label Single arm Dose escalation and expansion	Relapsed or refractory with ≥1 prior line of therapy, including an anti-CD20 therapy ECOG performance status 0–1 Life expectancy ≥ 6 months	145 patients with CD20+ B-cell malignancies (including 85 patients with DLBCL) Median age 67 years 70% male participants Median three prior lines of therapy
Melchardt et al., 2023 [33]	Glofitamab (CD20/CD3) in combination with polatuzumab vedotin	Phase II Multicenter Open label Single arm Feasibility trial	Treatment naïve ≥60 years and not eligible for full dose R-CHOP Allowed lymphoma related ECOG performance status ≤3 and reduced renal function	10 patients with DLBCL Median age 79 years
Hutchings et al., 2019 [34]	Glofitamab (CD20/CD3) in combination with atezolizumab	Phase Ib Multicenter Open label Single arm	Relapsed or refractory with ≥1 prior line of therapy, ineligible for or had exhausted standard therapeutic options * ECOG performance status 0–2 *	38 patients with B-NHL (including 33 patients with aggressive B-NHL) 52.6% male participants Median age 67 years Median three prior lines of therapy
Hutchings et al., 2023 [35]	Glofitamab (CD20/CD3) in combination with RO7227166	Phase I/II * Multicenter * Open label Single arm * Dose escalation	Relapsed or refractory with ≥2 prior lines of therapy, ineligible for or had exhausted standard therapeutic options * ECOG performance status 0–1 * Life expectancy ≥ 12 weeks *	104 patients enrolled, including 57 patients with LBCL Median age 63 years 61% male participants Median three prior lines of therapy
Dickinson et al., 2022 [36]	Glofitamab (CD20/CD3) as monotherapy	Phase II Multicenter Open label Single arm Dose escalation/expansion	Relapsed or refractory to ≥2 prior lines of therapy including anti-CD20 antibody and an anthracycline ECOG performance status 0–1 Life expectancy ≥ 12 weeks	154 patients with LBCL Median age 66 years 65% male participants Median three prior lines of therapy
Falchi et al., 2023 [37]	Epcoritamab (CD20/CD3) in combination with R-CHOP	Phase Ib/II Multicenter * Open label Dose escalation and expansion *	Treatment naïve CD20+ DLBCL IPI score ≥ 3	47 patients with DLBCL Median age 64 years
Vermaat et al., 2023 [38]	Epcoritamab (CD20/CD3) in combination with R-mini-CHOP	Phase Ib/II Multicenter Open label Dose escalation and expansion	Treatment naïve Not a candidate for full dose R-CHOP due to age ≥75 years or ≥65 years with comorbidities	28 patients with CD20+ DLBCL Median age 81 years
Brody et al., 2023 [39]	Epcoritamab (CD20/CD3) in combination with GemOx	Phase Ib/II Multicenter * Open label Dose escalation and expansion *	Relapsed/refractory disease Ineligible for HDT-ASCT *	34 patients with DLBCL Median age 71 years Median two prior lines of therapy
Abrisqueta et al., 2022 [40]	Epcoritamab (CD20/CD3) in combination with R-DHAX/C	Phase Ib/II Multicenter * Open label Dose escalation and expansion *	Relapsed/refractory CD20+ DLBCL Eligible for HDT-ASCT	29 patients with DLBCL Median age 58 years 72% of patients had received one prior line of therapy, 28% had received two or three prior therapies
Thieblemont et al., 2023 [41]	Epcoritamab (CD20/CD3) as monotherapy	Phase I/II Multicenter Open label Single arm Dose expansion	Relapsed or refractory to ≥2 prior lines of therapy including anti-CD20 antibody Failed or ineligible for ASCT ECOG performance status 0–2	157 patients with LBCL Median age 64 years 59.9% male participants Median three prior lines of therapy
Vose et al., 2023 [42]	Epcoritamab (CD20/CD3) as monotherapy	Phase IIa Multicenter Open label Single arm Dose optimization	Relapsed or refractory to ≥2 prior lines of therapy including an anti-CD20 monoclonal antibody containing regimen * Failed or ineligible for ASCT *	24 patients with CD20+ DLBCL Median age 65 years Median three prior lines of therapy (range 2–10)
Patel et al., 2022 [43]	Plamotamab (CD20/CD3) as monotherapy	Phase I Multicenter Open label Dose escalation	Relapsed or refractory disease, ineligible for or had exhausted standard therapeutic options ECOG performance status 0–2	36 patients with B-cell NHL (including 19 patients with DLBCL) Median age 67 years 61% male participants Median four prior lines of therapy
Song et al., 2023 [44]	GB261 (CD20/CD3) as monotherapy	Phase I/II Multicenter Open label Single arm Dose escalation and expansion	CD20+ r/r B-NHL or CLL with no available standard-of-care treatments ECOG performance status 0–1 *	47 patients with B-cell NHL (including 36 patients with DLBCL) Median age 60 years 55.3% male participants Median three prior lines of therapy
Olszewski et al., 2023 [45]	Mosunetuzumab (CD20/CD3) in combination with CHOP	Phase II Multicenter Open label Single arm Dose expansion	Treatment naïve IPI score 2–5 ECOG performance status 0–2 Life expectancy ≥ 24 weeks	40 patients with DLBCL Median age 65 years 55% male participants
Matasar et al., 2024 [46]	Mosunetuzumab (CD20/CD3) as monotherapy	Phase I/II Multicenter Open label Single arm Dose escalation	Relapsed or refractory to ≥2 prior lines including ≥1 prior anthracycline, and ≥1 anti-CD20-directed therapy) (DLBCL/tFL)	218 patients with B-cell NHL (including 102 patients with LBCL) Median age 64 years 66.5% male participants Median three prior lines of therapy
Budde et al., 2024 [47]	Mosunetuzumab (CD20/CD3) in combination with polatuzumab vedotin	Phase Ib/II Multicenter Multi-arm Open label Single arm Dose escalation and expansion	Relapsed or refractory to ≥1 prior line of therapy including anti-CD20 antibody Transplant ineligible ECOG performance status 0–2 Life expectancy ≥ 12 weeks	120 patients with B-cell NHL (including 117 patients with LBCL) Median age 68 years 67.5% male participants Median two prior lines of therapy

* Data taken from ClinicalTrials.gov (accessed on 26 May 2024). Abbreviations: HSCT, hematopoietic stem cell transplant; ECOG, Eastern Cooperative Oncology Group; DLBCL, diffuse large B-cell lymphoma; NHL, non-Hodgkin’s lymphoma; CAR, chimeric antigen receptor; TCE, T-cell engager; R-CHOP, rituximab/cyclophosphamide/doxorubicin/vincristine/prednisone; LBCL, large B-cell lymphoma; IPI, International Prognostic Index; GemOx, gemcitabine/oxaliplatin; R-DHAX/C, rituximab/dexamethasone/cytarabine/and oxaliplatin or carboplatin; HDT, high-dose therapy; CLL, chronic lymphocytic leukemia; tFL, transformed follicular lymphoma.

### 3.3. Treatment-Emergent Adverse Events

#### 3.3.1. Cytokine Release Syndrome (CRS)

There was considerable variability in the rates of CRS between all 19 studies, as presented in Table 2 [29,30,31,32,33,34,35,36,37,38,39,40,41,42,43,44,45,46,47]. The any grade CRS rates ranged from a low of 0% to a high of 72.2%. This variability was seen across both the monotherapy and combination trials, and both CD19/CD3 and CD20/CD3-targeting bsAbs. Grade ≥ 3 events were only reported in nine of these 18 studies, with rates ranging from 1.3% to 7% [32,35,36,37,38,39,41,46,47] (Appendix A).

#### 3.3.2. Immune Effector Cell-Associated Neurotoxicity Syndrome (ICANS)

The rates of ICANS were also variable across the 15 studies reporting data, but the events were largely infrequent and low grade [30,31,32,33,36,37,38,39,40,41,42,44,45,46,47]. The highest overall rate of 27% was reported in a CD19/CD3 bsAb [30], while median rates in the CD20/CD3 bsAbs ranged from 0% to 12% [32,33,36,37,38,39,40,41,42,44,45,46,47]. All of the studies reporting CRS and ICANS included mitigation strategies during the initial dosing period, such as step-up dosing, inpatient monitoring, and premedications (Appendix A). 

#### 3.3.3. Infections, Fever, and Fatigue

The rate of infection, fever, and fatigue were high amongst the studies reporting on these adverse events. Infection rates were reasonably consistent across the eight studies reporting this data. The median of any grade infection rates ranged from 41.6% to 49%, while the median grade ≥3 infection rates ranged from 14.6% to 23% [32,33,36,41,42,45,46,47]. Six of these studies reported recommending anti-infective prophylaxis for standard-risk patients, as per institutional practice [32,36,41,45,46,47]. 

There was heterogeneity in the fever rates over the 11 studies that reported this data, with the median of any grade fever rates ranging from 18.2% to 73% [29,32,34,35,36,37,41,43,45,46,47]. Only three of the studies reported grade ≥3 fever [29,32,46]. Moderate rates of fatigue were reported in 14 of the studies, with the median of any grade rates ranging from 17.9% to 46.7% [29,31,32,34,35,36,37,38,40,41,42,45,46,47]. The highest rates of any grade and grade ≥3 fatigue were reported by Olszewski et al. [45] and Budde et al. [47], who both investigated mosunetuzumab in combination with other therapies.

#### 3.3.4. Cytopenias

Cytopenias were reported across all 19 studies, with anemia, thrombocytopenia, and neutropenia most frequently reported (see Table 3). The rates of anemia were variable across the 18 studies reporting this data, with median any grade rates ranging from 4.4% to 44.5% and median grade ≥3 anemia rates ranging from 0% to 25% [29,30,31,32,33,34,35,36,37,38,39,40,41,42,43,45,46,47]. The thrombocytopenia rates also varied, with the median any grade events ranging from 11.5% to 40.7%, and median grade ≥3 events ranging from 5.7% to 17.4% [29,31,32,36,39,40,41,42,43,45,46,47]. The median rates of any grade neutropenia ranged from 4.4% to 36.5%, with the median rates of grade ≥3 neutropenia ranging from 14.6% to 25% [29,30,31,32,34,35,36,37,38,39,40,41,42,43,44,45,46,47]. The rates of leukopenia and lymphopenia were not widely reported. The combination study by Olszewski et al. [45] investigating mosunetuzumab with CHOP chemotherapy reported high rates of any grade anemia (42.5%), thrombocytopenia (25%), and neutropenia (70%). There were no clear differences between the CD19 and CD20-targeting antibodies.

#### 3.3.5. Other Frequently Reported Adverse Events

There was moderate heterogeneity in the reporting of other frequently occurring adverse events across the studies (Table 4 and Appendix A). Viardot et al. reported high rates of neurological symptoms, including any grade tremor (47.8%), grade ≥3 encephalopathy (8.7%), and grade ≥3 aphasia (8.7%) [29]. Bannerji et al., investigating single-agent odronextamab, reported high rates of any grade chills (47%) and hypophosphatemia (29% any grade, 19% grade ≥3) [32]. Pneumonia was the main infection reported across the studies, with grade ≥ 3 events reported by Viardot et al. (13%) [29], Bannerji et al. (9%) [32], and Olszewski et al. (7.5%) [45]. High rates of any grade diarrhea were reported across seven of the studies, ranging from 20.4% to 59% [29,34,39,41,43,45,47]. There were no clear differences in the type and rate of adverse events reported between the studies using bsAbs as monotherapy compared to combination therapy, or between the CD19 and CD20-targeting bispecific antibodies.

### 3.4. Efficacy Outcomes

Efficacy outcomes were reported by 16 of the included studies, including three studies as front-line therapy and 13 studies in relapsed/refractory disease, as summarized in Table 5 [29,30,31,32,34,35,36,37,38,39,40,41,43,44,45,46,47]. 

In the front-line setting, the ORR was high at 95% to 100% [37,38,45]. These were mostly complete responses, with a median CRR of 80.5% to 90% [37,38,45]. All three of these studies were investigating CD20/CD3 bsAbs in combination therapy [37,38,45]. 

In the relapsed/refractory setting, the responses were more variable. The reported median follow-up period ranged from 4.2 months to 17.9 months [29,32,36,40,41,44,46,47]. The median ORR ranged from 40% to 91%, and the median CRR ranged from 0% to 59% [29,30,31,32,34,35,36,39,40,41,43,44,46,47]. This variability was seen across both monotherapy and combination trials and CD19 and CD20-targeting antibodies. The highest individual ORR of 100% was reported by Abrisqueta et al. in their study of epcoritamab with chemotherapy, followed by autologous stem cell transplant [40]. The median DoR was reported in eight studies, ranging from 4.4 months to not reached [29,32,36,41,47]. The median PFS was reported in five studies, ranging from 3.7 months to not reached [29,30,32,36,40,41,44,47]. 

The method of measuring efficacy outcomes varied between the studies. The majority (12 studies) used the Lugano response criteria [32,34,35,36,37,38,39,40,41,44,45,47]. The remaining studies either used the Cheson response criteria [29,46] or did not report the response criteria used [31,43]. 

### 3.5. Quality Assessment

The quality of the 19 studies included in the analysis was assessed using the RoBANS 2 tool [28]. All of the studies provided a clear description of the trial design, participants enrolled, and the intervention [29,30,31,32,33,34,35,36,37,38,39,40,41,42,43,44,45,46,47]. 

For the safety outcomes, 10 of the studies were assessed to have low risk of bias across all eight risk-of-bias domains [29,30,31,32,36,41,42,45,46,47], as presented in Appendix B. The remaining nine studies were assessed to have unclear risk of bias, as they did not provide sufficient information on the grading criteria used for adverse event measurement [33,34,35,37,38,39,40,43,44]. The open-label design potentially increases the risk of bias for safety outcomes in domain 5 (blinding of assessors) for all 19 included studies; however, the selection of TEAE over TRAE was chosen to limit this risk. 

For the efficacy outcomes, six of the studies were assessed to have low risk across all eight risk-of-bias domains [29,30,36,37,41,47]. Four studies were assessed as high risk in domain 5 (blinding of assessors), as the efficacy outcomes were not independently assessed [32,34,45,46]. The remaining studies were unclear in terms of one or more domains [31,35,38,39,40,43,44] or provided no information [33,42]. 

All of the studies were limited by their non-randomized, open-label trial design without an active comparator. This meant that any comparisons to standard-of-care therapies were of a lower certainty of evidence due to variability in the study design, population, and method of outcome measurement. 

## 4. Discussion

The aim of this systematic review was to summarize the safety and efficacy of bsAbs in adults with LBCL, using data from across multiple clinical studies. Due to the increased use and availability of bsAbs in adults with LBCL, understanding the safety and efficacy is important in the management of these patients. This systematic review identified 19 interventional clinical trials for inclusion, published between 2016 and 2024, with nine different bsAbs investigated [29,30,31,32,33,34,35,36,37,38,39,40,41,42,43,44,45,46,47].

We demonstrate that the types of TEAEs from bsAbs in adults with LBCL generally aligns with their use in other hematologic malignancies; however, there exist differences in the incidence of TEAEs. Although significant infection rates were observed, the overall rates were lower than previously observed with bsAbs in multiple myeloma [22]. Similar to bsAbs in other settings [11,16,53], the incidence of CRS was high, but the rate of severe CRS was low. The trends were less clear for ICANS, due to heterogeneity in the study design and grading criteria used. One study reported higher rates of ICANS than expected (27%); however, this was a first-in-human study of AZD0486, where the recommended step-up dosing schedule was still being determined [30]. The study by Viardot et al., investigating blinatumomab, reported individual neurological events instead of ICANS [29], although many of these events were consistent with symptoms of ICANS [49]. High rates of neurological events have also been highly reported in studies of blinatumomab in acute lymphoblastic leukemia (ALL), with any grade rates ranging from 20 to 53% [54]. In contrast, Hawkes et al., who reported no cases of ICANS, were investigating a CD19/CD47 bsAb (TG-1801) [31], which does not engage the CD3 immune cells associated with ICANS [49]. The rates of fever, fatigue, and cytopenias were generally consistent with previous experience of bsAbs in other hematological malignancies [11,16,53], with no notable new trends observed. The high rates of fatigue are important given the impact of this toxicity on the patient’s quality of life [55]. However, evaluating the impact of this toxicity is limited by the lack of published patient-reported outcome data from bsAbs in lymphoma.

In comparing the adverse events described with bsAbs in this review to current standard-of-care therapies, there were some similarities, although the overall safety profile of bsAbs appeared unique. The high rates of infection and cytopenias observed are also commonly seen in many other therapies, including R-CHOP [56,57], salvage chemotherapy [58,59,60], antibody–drug conjugates [5], and monoclonal antibodies [6]. Some of the other adverse events frequently observed in bsAbs, including fatigue, fever, and peripheral neuropathy, are also associated with cytotoxic therapies [56,57,58,59] and antibody–drug conjugates [5]. The rates of fatigue and fever reported here are comparable to polatuzumab vedotin (PV), while the rates of peripheral neuropathy were reported as lower with bsAbs in this review [5]. The peripheral neuropathy observed with PV is due to the anti-microtubule agent conjugated to the antibody [60], while neuropathy in bispecific antibodies may be part of the neurotoxicity process triggered by systemic inflammation and cytokine release. Other adverse events commonly observed in standard-of-care therapies, such as cardiotoxicity, mucositis [56,57,58,59], or rashes [6], were not reported in the studies reviewed herein. This was an expected finding due to the unique mechanism of action of bsAbs, where toxicities are closely related to on-target off-tumor effects [8]. 

The most notable difference between the bsAbs reviewed here and standard-of-care therapies were the rates of CRS and ICANS. These toxicities are associated with treatments that target immune effector cells, so are primarily observed in patients exposed to bsAbs and CAR T-cell therapies [49]. The rates of CRS in this review ranged from 0% to 72.2%, in comparison to the pivotal CAR T-cell trials, where the overall rates of CRS were reported to range from 42% to 92% [61,62,63,64]. The incidence of ICANS was also lower in this review (ranging from 0 to 27%) compared to CAR T-cell therapies, where the rates ranged from 21% to 67% [61,62,63,64]. The lower rates of CRS and ICANS found may be partly due to the dosing schedule for bsAbs. While CAR T-cell therapies are administered as a single dose, most of the included bsAbs were administered using step-up dosing schedules in order to modulate the T-cell activation and cytokine release, thus mitigating severe toxicities. The difference in toxicity rates may also be due to the more highly activated nature of the manufactured T-cells in CAR T-cell therapies, resulting in increased inflammatory cytokine release and associated toxicities [65]. Along with lower rates of grade ≥3 cytopenias and infections, the bsAbs reviewed here appear less acutely toxic than CAR T-cell therapies in adults with LBCL. Long-term toxicities of CAR T-cell therapies in B-cell lymphomas include prolonged cytopenias and hypogammaglobulinemia [66]. Data on the long-term adverse effects of bsAbs are not yet available, but such effects may be similar or possibly more severe due to sustained B-cell depletion as a result of repeated dosing [67]. 

The efficacy outcomes of bsAbs reported in this review show promising responses in both the front-line and relapsed/refractory setting for patients with LBCL. The ORR reported for the bsAbs used in combination, in newly diagnosed patients, show superior outcomes to the reported ORR with traditional R-CHOP therapy, of 70% [68]. Responses to bsAbs in relapsed/refractory patients with LBCL also appear superior. The median ORR of bsAbs was reported as 40% to 91%, compared to approximately 26% with traditional salvage chemotherapy [69]. In contrast, comparisons to CAR T-cell therapy show inferior efficacy outcomes reported here, than in the pivotal CAR T-cell clinical trials. In relapsed/refractory patients, the ORR for lisocabtagene maraleucel, tisagenlecleucel, and axicabtagene ciloleucel have been reported as 52% to 82% [61,63,64,70]. 

Of the studies excluded from this review due to insufficient safety outcome data, the efficacy outcomes recently reported by Abramson et al. in June 2024 are noteworthy [71]. This phase III randomized trial (STARGLO) investigated glofitamab plus gemcitabine with oxaliplatin (Glofit-GemOx) versus rituximab-GemOx, in 274 patients with relapsed/refractory DLBCL. At an MFU of 20.7 months, the responses were superior in the Glofit-GemOx group, with a median overall survival of 25.5 versus 12.9 months, a median PFS of 13.8 versus 3.6 months, and a median CRR of 58.5% versus 25.3% [71]. We await the final published report for both the efficacy and TEAE and subsequent inclusion within our specified criteria. Nonetheless, the data from STARGLO support the notion of the efficacy of bsAbs in both the front-line and relapsed/refractory setting when given as monotherapy or in combination with chemotherapy. 

Two important questions arise for future empirical bedside and bench studies. The first is the utility, or lack thereof, of combining bispecific antibodies with chemotherapy. On one hand, the addition of chemotherapy may augment tumor clearance and promote a more immune-permissive microenvironment, thereby increasing the ORRs of bsAbs. On the other hand, chemotherapy may deplete important T-cell subsets that are co-opted by bsAbs. Further, bench-based correlative and comparative clinical trials will be needed to elucidate this important question. An additional, important question in the coming years will be the optimal sequencing of bsAbs and CAR T-cell therapies. Several of the included studies reviewed here found that bsAbs remained efficacious in patients previously exposed to CAR T-cell therapy [32,35,36,41,43,47]. A recent retrospective study of 47 patients with LBCL also found no impact on the efficacy of CAR T-cell therapies in patients previously exposed to bsAbs [72]. Without long-term efficacy data, it is not yet possible to confirm whether these observations will translate into positive overall survival outcomes. 

Overall, the results of this systematic review provide evidence that bsAbs are generally safe and effective in adults with LBCL. While the observed adverse events were consistent with expectations, the frequently occurring adverse events observed require careful consideration due to their potential impact on the morbidity and mortality of patients. The high infection rates found are important to emphasize, as prior research has shown infection is an independent risk factor for mortality in DLBCL patients [20]. Specialist care and inpatient management are likely to be required for the management of CRS, ICANS, and cytopenias [53,73]. Adverse event mitigation strategies, such as those used in the studies in this review, along with regular monitoring and early intervention, will be important in managing the safety of bsAbs in adults with LBCL. 

### 4.1. Limitations

There were several limitations in the study design and method of the outcome measurements used in the 19 studies included in this review. The heterogeneity in the study design and interventions precluded pooling of the studies, thus limiting the certainty of the observed effects. Although we undertook a broad search strategy for published and unpublished data, the reported data that may be within clinical trial registries were not included. Nonetheless, the targeted inclusion of trials assessing LBCL patients allows population-specific observations to be drawn. 

All the studies were non-randomized trials without active comparators, thereby limiting the certainty of any comparisons to standard-of-care therapies [74]. The open-label design may have also contributed to adverse event detection bias due to participants’ and investigators’ awareness of the treatment assignment [75]. Individual trial results may have been influenced by selection bias, with small sample sizes and targeted recruitment of participants with high functional status [76]. This also limits the generalizability of the results. Despite this, other demographic factors, such as age and gender, appear to be representative of the wider LBCL population [3]. 

Comparisons between the safety and efficacy in terms of bsAbs and standard-of-care therapies are further limited by the variability in outcome measurement methods used in the included studies. This was particularly true for ICANS, where only three of the studies reported using the currently recommended ASTCT consensus grading [49], meaning the overall ICANS rates may be underreported. Furthermore, the method of grading adverse events was unclear in nine of the studies. Also underreported, and thus not synthesized herein, was the timing, impact, and/or management of adverse events, an important aspect for the clinical implementation of these therapies. The overall impact of these limitations on the validity and accuracy of the results of this systematized review is unclear until results from randomized controlled trials (RCTs) using consensus grades are included.

### 4.2. Future Directions

There are several areas that would benefit from additional research to further describe the safety and efficacy of bsAbs in adults with LBCL. With RCTs currently underway for several of the bsAbs in adults with LBCL [77,78,79,80,81], this review of the literature and meta-analysis of the data should be repeated once these results become available. With larger sample sizes it would also be possible to perform subgroup analyses to investigate the differences in outcomes among different populations, such as age or gender groups. The impact of adverse events on the patient’s quality of life and the burden on the healthcare system should also be explored in more detail. Future research could incorporate patient-reported outcome data and invite consumer input to guide decisions on which adverse events are considered important. As access to drugs is often driven by health technology assessments, future research into the financial impact and burden on the healthcare system of managing the observed adverse events would be valuable. Longer term follow-up of the current clinical trials should be conducted to evaluate the impact of efficacy outcomes of bsAbs on overall survival trends. Overall, data on the safety and efficacy of bsAbs in adults with LBCL are still evolving and further research is required to gain a comprehensive understanding of this topic. 

## Figures and Tables

**Figure 1 ijms-25-09736-f001:**
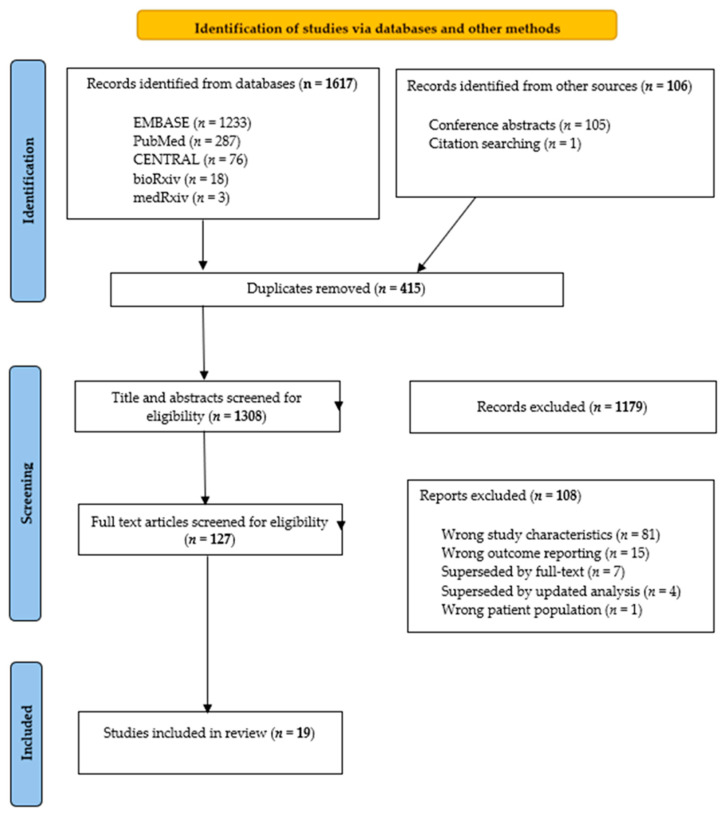
Preferred Reporting Items for Systematic Reviews and Meta-analysis (PRISMA) flowchart for study selection. Adapted from the PRISMA statement, 2020 [23]. *n* = number of studies.

**Table 2 ijms-25-09736-t002:** Median reported rates (%) of selected adverse events per bispecific antibody.

Bispecific Antibody	CRS		ICANS		Infection		Fever/Pyrexia		Fatigue	
	Any Grade	Grade ≥ 3	Any Grade	Grade ≥ 3	Any Grade	Grade ≥ 3	Any Grade	Grade ≥ 3	Any Grade	Grade ≥ 3
Blinatumomab [29]	0	0	NR	NR	NR	NR	43.5	4.3	26.1	0
AZD0486 [30]	48	0	27	9.7	NR	NR	NR	NR	NR	NR
TG-1801 [31,51]	0	0	0	0	NR	NR	NR	NR	25	0
Odronextamab [32]	61	7	12	3	49	23	73	1	33	5
Glofitamab [33,34,35,36]	50.2	2.6	4	1.5	49	17.5	18.2	0	17.9	0.6
Epcoritamab [37,38,39,40,41,42]	46.4	2.3	2.5	0	41.6	14.6	31.8	0	25	1.9
Plamotamab [43]	72.2	0	NR	NR	NR	NR	38.9	NR	NR	NR
GB261 [44]	12.8	0	0	0	NR	NR	NR	NR	NR	NR
Mosunetuzumab [45,46,47]	39.4	2.5	4.9	1.6	46.8	15.1	20	0.9	46.7	6.7

Abbreviations: CRS, cytokine release syndrome; ICANS, immune effector cell-associated neurotoxicity syndrome; NR, not reported.

**Table 3 ijms-25-09736-t003:** Median reported rates (%) of cytopenias per bispecific antibody.

Bispecific Antibody	Anemia		Thrombocytopenia		Neutropenia		Leukopenia		Lymphopenia	
	Any Grade	Grade ≥ 3	Any Grade	Grade ≥ 3	Any Grade	Grade ≥ 3	Any Grade	Grade ≥ 3	Any Grade	Grade ≥ 3
Blinatumomab [29]	4.4 *	NR	21.7	17.4	4.4	NR	17.4	17.4	NR	NR
AZD0846 [30]	34	0	NR	NR	32	NR	20.9	NR	29	NR
TG-1801 [31,51]	31	6	25	13	19	19	NR	NR	NR	NR
Odronextamab [32]	38	25	28	14	25	19	19	9	22	19
Glofitamab [33,34,35,36]	25.8	6	25	8	28.6	22.5	NR	NR	NR	3.2
Epcoritamab [37,38,39,40,41,42]	33	10.2	40.7	5.7	36.5	14.6	NR	NR	NR	NR
Plamotamab [43]	44.5	19.4	NR	11.1	NR	16.7	NR	NR	NR	NR
GB261 [44]	NR	NR	NR	NR	31.9	17	NR	NR	NR	NR
Mosunetuzumab [45,46,47]	15.1	19.2	11.5	12.2	35	25	7.5	7.5	8.4	10

* Data taken from ClinicalTrials.gov (accessed on 26 May 2024). Abbreviations: NR, not reported.

**Table 4 ijms-25-09736-t004:** Other frequently reported adverse events per bispecific antibody.

Bispecific Antibody	Any Grade, Occurring in >20% of Participants		Grade ≥ 3, Occurring in >5% of Participants	
	Adverse Event	Median Rate, %	Adverse Event	Median Rate, %
Blinatumomab [29]	Tremor	47.8	Device-related infection	13
	Oedema	26.1	Pneumonia	13
	Device-related infection	21.7	C-reactive protein increased	13
	Pneumonia	21.7	Encephalopathy	8.7
	Diarrhea	21.7	Aphasia	8.7
			Hyperglycemia	8.7
AZD0486 [30]	N/A	N/A	NR	NR
TG-1801 [31,51]	Headache	25	Fall	6
	Abdominal pain	25		
	COVID-19	25		
Odronextamab [32]	Chills	47	Hypophosphatemia	19
	Hypophosphataemia	29	Aspartate transaminase increased	10
	C-reactive protein increased	28	Pneumonia	9
	Cough	28	Hypotension	8
	Hypotension	28	Alanine transaminase increased	7
	Headache	25	Hypoxia	6
	Nausea	24	Hyperglycemia	6
	Infusion-related reaction	24		
	Decreased appetite	23		
	Dyspnea	22		
	Blood creatinine increased	21		
	Tachycardia	21		
	Oedema peripheral	21		
Glofitamab [33,34,35,36]	COVID-19	25	Bleeding	10
	Diarrhea	21	Renal toxicity	10
	Decreased appetite	21	Hypophosphatemia	5.8
Epcoritamab [37,38,39,40,41,42]	Diarrhea	39.7	NR	NR
	Nausea	34		
	COVID-19	32		
	Injection site reactions	28		
	Constipation	21		
	Hypokalema	21		
Plamotamab [43]	Nausea	38.9	NR	NR
	Asthenia	27.8		
	Diarrhea	25		
	Hypophosphatemia	25		
	Aspartate transaminase increased	25		
GB261 [44]	COVID-19 infection	40.4	COVID-19 infection	12.8
Mosunetuzumab [45,46,47]	Nausea	42.5	Febrile neutropenia	20
	Constipation	37.5	Hypophosphatemia	14.7
	Hypokalemia	32.5	Pneumonia	7.5
	Peripheral neuropathy	31.7	Hypokalemia	7.5
	Diarrhea	31.7	Decreased appetite	7.5
	Decreased appetite	30		
	Vomiting	30		
	Alopecia	30		
	Dizziness	25		
	Headache	23.4		
	Hypotension	22.5		
	Hypophosphatemia	22.5		

Abbreviations: N/A, not applicable; NR, not reported.

**Table 5 ijms-25-09736-t005:** Median reported efficacy outcomes per bispecific antibody.

Bispecific Antibody	Sample Size	MFU, Months	ORR, %	CRR, %	PRR, %	mDoR, Months	mPFS, Months	Response Criteria
Front-line setting								
Epcoritamab [37,38]	*n* = 66	11.5	100	80.5	NR	Not reached	Not reached	Lugano
Mosunetuzumab [45]	*n* = 40	32	95	90	5	Not reached	NR	Lugano
Relapsed/refractory setting								
Blinatumomab [29]	*n* = 20	15	40	20	20	11.6	3.7	Cheson
TG-1801 [31]	*n* = 9	NR	56	0	56	NR	NR	NR
Odronextamab [32]	*n* = 82	4.2	36	24	11.5	4.4	6.8	Lugano
Glofitamab [34,35,36]	*n* = 243	12.6	52	39	NR	18.4	4.9	Lugano
Epcoritamab [39,40,41]	*n* = 206	10	91	59	22.1	12	4.4	Lugano
Plamotamab [43]	*n* = 19	NR	47.4	26.3	NR	NR	NR	NR
GB261 [44]	*n* = 22	4.5	73	45.5	NR	Not reached	NR	Lugano and LYRIC 2016
Mosunetuzumab [46,47,52]	*n* = 227	17.9	47	32.7	15.5	20.8	11.4	Cheson and Lugano

Abbreviations: *n*, number (of participants); MFU, median follow-up; ORR, overall response rate; CRR, complete response rate; PRR, partial response rate; mDoR, median duration of the response; mPFS, median progression-free survival; NR, not reported.

## Data Availability

No new data were created or analyzed in this study. Data sharing is not applicable to this article.

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
