# Peer review of "Safety and Efficacy of Bispecific Antibodies in Adults with Large B-Cell Lymphomas: A Systematic Review of Clinical Trial Data"

_ijms, 2024, doi:10.3390/ijms25179736_

Round 1

Reviewer 1 Report

Comments and Suggestions for Authors

In this review manuscript, Elena Bayly-McCredie and colleagues described the safety and efficacy of bispecific antibodies in adults with large B-cell lymphomas from clinical trial data. This study is novel and well-written; I have some minor comments. 

Line 51: This sentence is not clear. It is better to explain the structure of a bispecific antibody in the introduction; one arm of the Fab part binds to one antigen, and the other arm of the Fab part binds to a different antigen. The Fc part functions in an immune response by binding with immune-effector cells. 

Table 1 contains a lot of information; please clearly write what is in the bracket you are referring to. Also, you can include a hyperlink to each clinical trial from ClinicalTrials.gov. 

There are many typo mistakes throughout the manuscript, like an extra period in line 210, no period after 'Viardot et al' (line # 333), etc. 

Author Response

We thank Reviewer 1 for the time and effort in editing the manuscript              

  1. Line 51: This sentence is not clear. It is better to explain the structure of a bispecific antibody in the introduction; one arm of the Fab part binds to one antigen, and the other arm of the Fab part binds to a different antigen. The Fc part functions in an immune response by binding with immune-effector cells.

We agree that a succinct definition of bispecific antibodies is difficult in a field where the permutations and combinations of protein engineering are manifold and dynamic. To account for this, we have included the reviewer’s suggestion regarding antigen binding domains. However, because not all bispecific antibodies included in this review contain a Fab and Fc region (e.g. blinatumomab lacks an Fc region), we have not mentioned the Fc region. The updated definition now reads:

“By binding with immune-effector cells with one antigen-binding region and the tumor associated antigen with the other antigen-binding region, bsAbs promote immune-mediated immune cell activation and tumor cell death[8].”

  1. Table 1 contains a lot of information; please clearly write what is in the bracket you are referring to. Also, you can include a hyperlink to each clinical trial from ClinicalTrials.gov. 

Thank you, some of the less critical information has been removed including the location of the sites and the range of age and prior lines of therapy. The ClinicalTrials.gov study details have now been linked to each row.

  1. There are many typo mistakes throughout the manuscript, like an extra period in line 210, no period after 'Viardot et al' (line # 333), etc. 

Thank you, typos have been corrected including adding a period after all in text citations of “et al”.

Reviewer 2 Report

Comments and Suggestions for Authors

In this study, the authors systematically summarize the collective reports on efficacy and adverse events of bispecific antibody treatment of large B-cell lymphoma in adults. A systematized search over three databases was performed, an included only studies of sufficient quality according to various criteria. The study was conducted according to PRISMA (Preferred Reporting Items for Systematic reviews and Meta-Analyses) 2020 guidelines. Nineteen studies of nine bispecific antibodies, either as single agents or in combination therapies, were finally included. The report concludes that the bispecific antibodies are efficient judging from the ORR (95–100% in the front-line and 36–91% in R/R disease), and generally well tolerated. In comparison with CAR-T-based therapies, BsAbs are of lower efficacy, but better tolerable in adults with LBCL. The study is very well designed, all parameters used for evaluation of literature data are well defined, and the limitations of the study are outlined as well. The large body of data is well presented, and the report is interesting to read.

Please find below a list of minor remarks:

Line 162: 3.2. Study characteristics – wrong formatting, should be a title

Line 177: “in treatment naïve patients“ – in treatment of naïve patients

Lines 192-193: propose to change the order of subsentences: “Seven studies, all of which were conference abstracts, did not report how adverse events were graded”

Line 224. “Median any grade infection rates” – median of any grade infection rates

Line 229: “median any grade fever rates” - median of any grade fever rates

Line 230: “any grade ≥3 fever“: grade ≥3 fever

Line 231: “median any grade rates ranging” – median of any grade rates

Line 304: “remaining nine studies by were assessed” - remaining nine studies were assessed

Line 344: There are certainly published data on side effects of Blinatumomab (also patient-reported), including fatigue, at least for ALL

Line 363:” so are primarily seen in bsAbs and CAR T-cell therapies“ – subsentence not completely clear

Line 366:” incidence of ICANS were also lower in this review” – it would be informative to summarize the numbers

Line 378:” depletion from repeated dosing”- depletion resulting from repeated dosing, and please include a reference

Author Response

Reviewer Two:

We thank reviewer 2 for the time and effort in editing and improving the quality and legibility of the manuscript.  

  1. Line 162: 3.2. Study characteristics – wrong formatting, should be a title.

Corrected

  1. Line 177: “in treatment naïve patients“ – in treatment of naïve patients.

We agree that this is often confusing terminology. On review of the existing literature we have taken note of the hyphen between treatment and naïve and this has been corrected. “In treatment of naïve patients” would, in our interpretation, would indicate that they patients are naïve and not naïve to treatment.

  1. Lines 192-193: propose to change the order of subsentences: “Seven studies, all of which were conference abstracts, did not report how adverse events were graded”

Agreed, thank you this has been updated.

  1. Line 224. “Median any grade infection rates” – median of any grade infection rates.

Updated as suggested.

  1. Line 229: “median any grade fever rates” - median of any grade fever rates.

Updated as suggested.

  1. Line 230: “any grade ≥3 fever“: grade ≥3 fever.

Updated as suggested.

  1. Line 231: “median any grade rates ranging” – median of any grade rates.

Updated as suggested.

  1. Line 304: “remaining nine studies by were assessed” - remaining nine studies were assessed.

Thank you, this has been corrected.

  1. Line 344: There are certainly published data on side effects of Blinatumomab (also patient-reported), including fatigue, at least for ALL.

We agree that the adverse event data on blinatumomab colours the interpretation of the data in LBCL, we state within our manuscript:

“High rates of neurological events have also been highly reported in studies of blinatumomab in acute lymphoblastic leukemia (ALL), with any grade rates ranging from 20-53%[53].”

We have also made note of the lack of patient-reported data in lymphoma specifically given this was the emphasis of our review , rather than generally across any cancer type.

  1. Line 363:” so are primarily seen in bsAbs and CAR T-cell therapies“ – subsentence not completely clear. Sentence updated
  2. Line 366:” incidence of ICANS were also lower in this review” – it would be informative to summarize the numbers.

Thank you, we agree and have added ranges.

  1. Line 378:” depletion from repeated dosing”- depletion resulting from repeated dosing, and please include a reference.

We apologise for the oversight and have updated the text and added an appropriate reference. The text now reads:

“Data on the long-term adverse effects of bsAbs is not yet available, but may be similar or possibly more severe due to sustained B-cell depletion resulting from repeated dosing[66].”  

Reviewer 3 Report

Comments and Suggestions for Authors

The authors searched and reviewed the safety and efficacy of bispecific antibodies in DLBCL patients. It is very interesting, but have some issues for publications.

1. Though interesting, but the conclusion has no novel findings,

2. The subjects are to heterogenous for the conclusion, such as different antibodies, different treatment protocol (monotherapy vs combination therapy), different phase, and different line (first line vs late status)

Author Response

  1. Though interesting, but the conclusion has no novel findings,

We thank the reviewer for their considered approach to our review. We wish to emphasize that this manuscript is a review and thus all findings would be, by the nature of a review, not novel. However, the review itself is indeed novel and the approach is novel. This is the first systematic review that we are aware of that specifically addresses the effect of bispecific antibodies in large B cell lymphoma. In addition, we assessed 1,617 abstracts from both published literature and conference abstracts. We also individually excluded records from which there was multiple reports of the same patient dataset. This detailed in our PRISMA flowchart in figure 1. This is particularly important in the modern reporting environment where they may be multiple reports of the same dataset. We have updated our Methods to highlight this:

“By sorting each individual study by clinical trials identification number, only those studies that reported on the largest, most comprehensive and recent results from the same dataset were included.”

We also provide, for the first time that we are aware of, a comprehensive table that details all bispecific antibodies in the published and grey literature in Tables 2-5a. We believe this to be an invaluable resource for the audience.

We further provide conclusions not only based on the clinical data, but coloured by our experience as bench-based scientists, as per the Discussion:

“Two important questions arise for future empirical bedside and bench studies. The first is the utility or lack thereof combining bispecific antibodies with chemotherapy. On one hand, the addition of chemotherapy may augment tumor clearance and promote a more immune-permissive microenvironment thereby ORRs from bsAbs. On the other hand, chemotherapy may deplete important T cell subsets that are co-opted by bsAbs. Further, bench-based correlative and comparative clinical trials will be needed to eluci-date this important question. An additional, important question in the coming years will be the optimal sequencing of bsAbs and CAR T-cell therapies. Several of the included studies reviewed here found that bsAbs remained efficacious in patients previously ex-posed to CAR T-cell therapy[30, 33, 34, 39, 41, 45]. A recent retrospective study of 47 patients with LBCL also found no impact on the efficacy of CAR T-cell therapies in patients previously exposed to bsAbs[71]. Without long-term efficacy data, it is not yet possible to confirm if these observations will translate to positive overall survival outcomes.”

  1. The subjects are to heterogenous for the conclusion, such as different antibodies, different treatment protocol (monotherapy vs combination therapy), different phase, and different line (first line vs late status)

We agree with the reviewer that the subjects in this study are indeed heterogeneous. However, because of the novelty of this analysis and the burgeoning set of literature in the field we felt it still necessary to present this work provide an overview of the field where it currently stands and avenues to future investigation. Yet, it is because of the heterogeneity outlined by the reviewer, that we opted not to undertake a meta-analysis but limit our manuscript to a systematic review. As specifically state in the abstract:

“Due to heterogeneity of the included studies, results were discussed as a narrative synthesis.”

As per Cochrane guidelines 9.5.3. we opted not to exclude studies but instead pursue a narrative synthesis. Nonetheless, we also still provide median figures of response adverse events and ranges in a readily appreciable format in the tables within the text.

Furthermore, to account for the different antibodies and treatment protocols and lines of therapy we have divided our tables, as per the reviewer, with Tables 5a and 5b reporting median reported efficacy outcomes per bispecific antibody in front-line and relapsed/refractory treatment.

In sum, our manuscript provides the reader with a comprehensive view of the landscape when treating LBCL with bispecific antibodies while accounting for the fact that this is a nascent and dynamic field.